# Measuring the Semi-Century Ecosystem-Service Value Variation in Mekelle City Region, Northern Ethiopia

**Shishay Kiros Weldegebriel** [1,*] and **Kumelachew Yeshitela** [2]

1   Department of Environment and Climate Changes Management, College of Urban Development and Engineering, Ethiopian Civil Service University, P.O. Box 5648, Addis Ababa 1000, Ethiopia
2   Ethiopian Institute of Architecture, Building Construction and City Development, Addis Ababa University, P.O. Box 511, Addis Ababa 1000, Ethiopia; kumelachew.yeshitela@eiabc.edu.et
*   Correspondence: shishaykiros@gmail.com

**Abstract:** The Mekelle city region is facing severe ecosystem degradation. The study area has experienced unprecedented land-use dynamics over the past 47 years, but the effect of these dynamics on ecosystem-service values remains unknown. Estimating the various ecosystem services from a city region perspective has not been attempted so far. The rationale of this study was to estimate the spatial–temporal ecosystem-service value variations. The methodology employed was land-use/land-cover (LULC) datasets of remotely sensed datasets of the years 1972, 1984, 2001, 2012, and 2019, and ecosystem service value coefficient, expert focus group discussion, and document review were used. The digital satellite images were processed, classified, and analyzed using Earth Resource Development Assessment System (ERDAS) Imagine. Computations of changes in the land-use categories were made using Arc GIS 10.5.1, Eviews for time series data analysis, and XLSTAT analytical tools were used. Over the whole study period from 1972 to 2019, a loss of USD 128.6 million was observed, which is a reduction of 501.9%. The study shows that due to land-use changes, the total ecosystem service value is decreasing annually, suggesting that much more severe ecosystem degradation is due to occur. The results are relevant to policy development and indicate that ecological restoration is the best option in the study area.

**Keywords:** city region; ecosystem service; land-use dynamics; valuation

## 1. Introduction

Land-use change has a substantial impact on the world's ecosystems. Changes in the extent and composition of ecosystems have large impacts on the provision of ecosystem services [1]. Land-use/land-cover (LULC) dynamics alter ecosystem-service values [2]. The value of ecosystem services is now widely acknowledged for their hopeful role in economic, environmental, and social well-being and in achieving the three main pillars of sustainable development [3]. Assessing the influence of LULC changes on ESV is an important tool to support decision-making [4] Ecosystem-service values (ESVs) provide an integrated, universal measure for evaluating and communicating the impacts of land-use dynamics and for justifying, prioritizing, and targeting investment in conservation [5].

Ecosystem-service valuation is the process of assessing the contributions of ecosystem services to sustainable human well-being [6]. Ecosystem services offer the potential to link environmental degradation and loss, matters of which are becoming increasingly prominent, with the sphere of economics and development. Ecosystem Services draw from both the current state of knowledge of ecology and economics and adopt the "received wisdom" from both in terms of their respective mechanisms and modes of operation [7]. The values or importance that people place on ecosystems were identified as a crucial dimension of sustainable management of social–ecological systems.

Losses in ecosystem services have direct economic consequences that we systematically underestimate. Making the value of our natural capital visible to economies and

society creates an evidence base to pave the way for more targeted and cost-effective solutions [8]. Valuation is often useful because many decisions involve trade-offs between ranges of things that affect human wellbeing differently. In these cases, we do not have a choice [9]. We can choose to make these valuations explicit or not; we can do them with an explicit acknowledgment of the huge uncertainties involved or not; but as long as we are forced to make choices, we are going through the process of valuation [10].

The environmental degradation related to LULC change costs Ethiopia about USD 4.3 billion per year [11] The last few decades have witnessed great change in policies related to land use in northern Ethiopia [12]. Nevertheless, a detailed study on the effects of these changes on ecosystem services value from the city region remains scanty. Quantitative and empirical information about land-use dynamics and ecosystem-service values is limited. The current situation in the Mekelle city region indicates that there is a lack of information, understanding, and planning about the effects of LULC changes which is leading to the loss of essential and beneficial ecosystem services.

The empirical results about the scientific basis for integrating ecosystem services into land-use decisions are still lacking. Some key variables regarding the problem have been overlooked. Several efforts have been made to improve the quantification of ecosystem services and to understand ecosystems' contribution to human well-being have been made. From the reviewed works of literature, there are still deficiencies and methodological incon- sistencies. As per the literature review, studies made about ESV in the Mekelle city region are scarce, and the understanding of the supply and threats of the ecosystem services in the study area is unclear. Therefore, the objectives of this study are to: quantify the spatial and temporal variations of ecosystem-service values, quantify the critical ecosystem services that have had their values affected by land-use dynamics, and analyze the relationship between ecosystem-service values and land-use dynamics.

## 2. Materials and Methods

### 2.1. Study Area

The study area is located within the Tigray region, the northern part of Ethiopia found in the west 39.362942, east 39.687048, north 13.680920 and south 13.342621 about 760.61 km north of Addis Ababa and the area covered in this investigation is 897.12 km$^2$ (Figure 1).

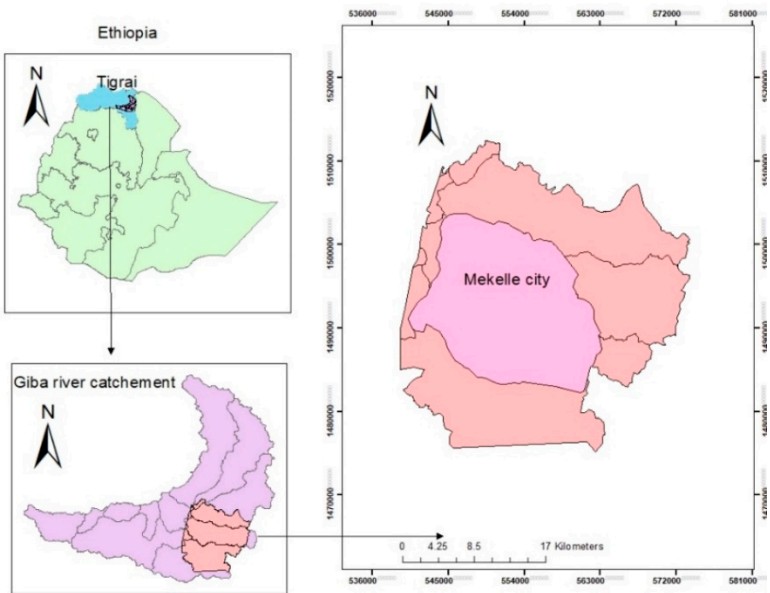

**Figure 1.** Location map of the study area.

In the densely populated study area, six major watersheds, i.e., Middle Geba, Agulae, May Gebat Ilala, some parts of Suluh and Genefel which are tributaries to the Giba river

basin—are the main water supply and sources of other ecosystem services for Mekelle metropolitan city. The study area is located within the Giba river catchment which is one of the biggest tributaries of the Tekeze river basin and has an area of 4019 km$^2$ [13] There is an increasing reliance on distant catchment sources to meet requirements such as freshwater, food, fresh air and other ecosystem services.

The study area is characterized by varied topographic conditions. The elevation ranges from 1700 in the Geba river to 2685 in the Ellala and Gebat river catchment. The core city of the study area, Mekelle city, is 2062 m above sea level. In the eastern part of the study area, the Aragure slopes have an elevation of 2400 to 2600 m above sea level and consist of steep slopes, descending into the Afar lowlands. The climate is predominantly semi-arid with irregular rainfall and frequent drought periods. The mean annual rainfall is estimated to be less than 532 mm [14]. Climatically, the area has a semi-arid climate with little variation and it is known for its environmental vulnerability [15]. The agro-climatic zone of the study area is a mild climatic condition. The monthly mean minimum temperature is 15 °C and the maximum monthly temperature may go as high as 28 °C. In general, two distinct seasons can be recognized in the Mekelle region. The first is the main rainy monsoon season which lasts from June to September, where most of the annual rainfall occurs. The second is the dry winter season from March to April. The study area population is 556,127 [16]. Similar to other arid and semiarid regions, most rivers and springs in the study area significantly decrease their discharge to unreliable levels (for domestic and small-scale irrigation water supply) starting from the early or mid-months of the dry season, and usually, they become dry in the late months of the dry season [17].

### 2.2. Data Sources

This study used an explanatory sequential mixed-method design. The combination of quantitative and qualitative approaches provides a more complete understanding of a research problem than either approach alone. First, quantitative research was collected and analyzed, and to validate the results of quantitative data in more detail, qualitative data were employed. This design involved a two-phase data collection in which the quantitative data were collected in the first phase, analyzes the results, and then uses the results to plan and build on to the second qualitative data phase.

#### 2.2.1. Land-Use/Land-Cover Change Detections

This landscape-scale level study employs a combination of analysis of satellite imageries (Table 1) of 47 years and information from field studies, document review, and key informant interview. To detect LULC dynamics, data from the Landsat series were extracted which are found at https://earthexplorer.usgs.gov/and NASA (accessed on 25 February 2019).

The years for analysis were selected based on key signs of LULC change, e.g., land degradation, land policy changes, rapid urbanization, rapid population growth, socioeconomic development, and finally the availability of satellite images. Images from the same period (November–January), i.e., immediately after the rainy season, were selected to minimize the seasonal effect on the classification results. From the above table, the Landsat series images have a medium spatial.

#### Land-Use/Land Cover Classification

The classification used in this study was heavily determined by the spatial resolution of the Landsat series, the classification system used, and author knowledge of the study area. The parameters used for land-use/land-cover classification are based on a comprehensive review of the literature. Based on field observation and general historical information gained from participants during the survey, it was decided to focus on the major land-use/land-cover classes presented in Table 2.

**Table 1.** Specifications of satellite images that were utilized in this study.

| Satellite | Sensor | Path | Row | Resolution (m) | Acquisition Date |
|---|---|---|---|---|---|
| Landsat 1 | MSS Multispectral C1 level 1 | 181 | 50 | 60 | 2 November 1972 |
| Landsat 5 | TM Multispectral C1 level 1 | 169 | 51 | 30 | 22 November 1984 |
| Landsat 5 | TM Multispectral C1 level 1 | 168 | 51 | 30 | 21 November 1992 |
| Landsat 5 | TM Multispectral C1 level 1 | 168 | 51 | 30 | 30 November 2001 |
| Landsat 7 | ETM$^+$ Multispectral C1 level 1 | 168 | 51 | 30 | 20 November 2012 |
| Landsat 8 | Operational Land Imager OLI/TIRS C1 level 1 | 169 | 51 | 30 | 23 January 2019 |

**Table 2.** Description of land-use/land cover classification.

| LULC Class | Description |
|---|---|
| Built-up area | The built-up area encompasses a developed area, i.e., land on which buildings and/or non-building structures are present, possibly as part of a larger developed environment such as the urban area of Mekelle city and its suburban area, developed land lot including roads, rural area, small towns, villages, residential, commercial and service, industrial, socioeconomic infrastructure, and transport units and mixed urban and other urban, transportation, roads including cross country roads and both urban and rural roads, and airports. |
| Grassland | Land under grass cover but highly managed for grazing and feeding of domestic animals, selling for various purposes. A land is conquered by natural grass, small herbs, and grazing lands. Those land units allocated as a source of animal feed, including communally owned grazing areas and also those owned by various institutions including churches and schools. |
| Cultivated land | Areas of land prepared for growing crops. This category includes areas currently under crop, irrigated agricultural land, and land under preparation/fallow land. Areas covered with annual and perennial crops covering; include most flat areas and also some steep slopes where various food crops are grown, either on a rain-fed basis or using irrigation. Irrigation is commonly practiced near rivers and streams, from dams, ponds, and using underground waters shades of Mekelle city region. |
| Bushes and Shrubs | Land covered with shrubs, bushes, and small trees with little woody vegetation mixed. A specific area is characterized by scattered bushes and shrubs and trees. The category includes areas covered with shrubs and bushes found in hilly areas. Trees and shrubs are classified based on height. Areas covered by low woody of 3 m in height, multiple stems, vertical growing of bushes, and shrubs with canopy cover between 5 and 50%. Examples include Acacia. Currently, the dominant one is *Euclea shimperi* locally known as Kilio. The cover types are in the average range between 1m and 3m in height and with a canopy cover of between 50 and 80%. In February, it also consists of dry grasses with some green herbaceous species such as *Rumex abyssinicus* locally known as "Hohot", *Aloe barbadensis* and *Ocimun lamiifolium.* |
| Waterbody | Any types of surface water or all areas of open water areas covered with water either along the river bed or man-made earth dams, filled sand dams and ponds including rivers, streams, swales, lakes and wetlands, Rivers, permanent/perennial and Intermittent rivers, and streams, open water, lakes, ponds, reservoirs |
| Natural Forest land | High and dense natural forest and in churches and reserved Areas. The dense forest is a forest thick with trees or having trees growing very closely. Examples include *Olea europaea* subsp. *cuspidata Ficus vasta, Ficus sur, Junuiperus prcserea, Dodonaea viscosa, Pruus africana,* and *Salix subserata* <br> This area was previously densely forested and dominated by *Juniperus procera* and *Olea europaea* subsp. *cuspidata* |

**Table 2.** *Cont.*

| LULC Class | Description |
|---|---|
| Bare lands | land left without vegetation cover (devoid vegetation), eroded land due to land degradation and weathered road surface and includes rock, including exposed soils, stock quarry, rocks, and areas of active excavation, vacant land within Mekelle city |
| Plantation Forest | Areas covered by man-made trees including sparse forests Examples include eucalyptus tree, acacia, and teak |
| Riverside vegetation | Plants growing in areas adjacent to rivers and streams in both urban and rural (Riparian vegetation includes trees, shrubs, grasses, and trees with canopies). |

The major steps using satellite imageries analysis were layer stack, band combination; study area rectangular preparation using ArcMap 10.5 to cut the large satellite image obtained using the path/row. For Landsat 7 fixing, Landsat scanline error/malfunction was employed to fill the gap of ETM plus using ArcGIS 10.5. A mask layer was employed so that LULC analysis was limited only to the study area of the research using ArcGIS spatial analyst tool extraction by mask was used and raster mask using the boundary of the Mekelle city region was generated.

Radiometric Adjustment

Contrast stretching, which adjusts colors, was applied to the selected satellite imageries using Earth Resource Development Assessment System (ERDAS) Imagine 2015.

Image Pre-Processing

Geometric correction procedure used to register each pixel that was generated to real-world coordinates. The images were geometrically corrected to the local coordinate system—Traverse Mercator projection using ERDAS Imagine.

Radiometric Correction

Dealing with multi-date image datasets requires that images gained by sensors of different times are comparable in terms of radiometric characteristics. Radiometric correction techniques such as image enhancement, normalization, and calibration were applied to multi-date satellite images to increase visual discriminations between features and increase the amount of information to improve interpretability and finally the images were being subset to fit the study area using ERDAS Imagine.

Supervised Image Classification

Supervised image classification techniques which are guided by the author were employed to prepare LULC maps. The supervised classification in this study was based on the selected sample pixels in imagery that are representative of specific classes identified and then direct the image processing software to use these training sites as references for the classification. Training sites are selected based on knowledge of the Mekelle city region. Then, the bounds are set for how similar other pixels must be to group them. These bounds are regularly set based on the spectral characteristics of the training area, plus or minus a certain increment based on brightness and strength of reflection in specific spectral bands.

Maximum likelihood classification (MLC) is based on the probability that a pixel that belongs to a particular class was chosen to classify LULC. Supervised image classification was carried out using the Maximum Likelihood (ML) classifier. The LULC classes for the study area were classified into 9 heterogeneous land cover classes. Ground verification was applied to verify and evaluate the accuracy of supervised classification. A total of 450 field points were collected using systematic random sampling. To get accurate LULC classification, the study made a reconnaissance visit to the study area. Further, ERDAS imagine AOI synchronized with online Google earth was used to identify each pixel and accurately prepare LULC. Moreover, signature editor signatures that are representative

for pixel were prepared. Band combinations of 4-3-2 near-infrared (NIR)-red-green for the thematic mapper images and 5-4-3 NIR-red-green) for the operation land imager image were used to make false-color composite images. In supervised image classification, training sample points were required to generate spectral signatures for each LULC type. For classifiers like the ML, a training sample size for each class was 10–30 times the number of bands for each LULC type. The accuracy assessment for each LULC types. is presented in Table 3.

The overall Kappa Statistic (Table 4) is 0.9164. The Kappa coefficient was generated from a statistical test to evaluate the accuracy of classification. The Kappa assessed how well the classification was performed. A value close to 1 show that the classification is better than random. In addition to this, accuracy might be low due to errors in reference data and errors in the classified map. To solve those problems, the study incorporates atmospheric correction, image enhancement, and employs different sensors.

### 2.2.2. Spatio-Temporal Land-Use Dynamics and Its Effects on Ecosystem-Service Values

The ESV analysis follows a mixed-methods approach, where the qualitative methodology is used to inform quantitative research. The quantitative part applied benefit-transfer methodology to monetarily evaluate the relevant ecosystem services. Focus group meetings were conducted with experts who have experience in landscape management. The experts described several ecosystem services to be quantified. A qualitative approach to the valuation of ecosystem services was used here as a supplementary to the results of the quantitative approach. Alone, the qualitative methods are useful when possible, changes need to be made. The qualitative methods were used to inform, validate and assess the monetary valuation made in the initial state of the quantitative result. Despite the output being a monetary value of ecosystem services, identification and involvement of relevant experts were important.

In the study area and other similar locations, no data exist about previously-carried-out evaluations of similar services. The LULC types with their area in hectares for the years 1972, 1984, 1992, 2001, 2012, and 2019 and biome counterparts with the corresponding value coefficients (USD $ha^{-1}$ $year^{-1}$) using global value coefficients adopted from [10] and modified conservative value coefficients mainly based on the Economics of Ecosystem and Biodiversity (TEEB) database by [18] were utilized.

In this research study, each LULC type of different reference years was compared with those representative biomes or LULC types to obtain their corresponding ecosystem service value coefficients identified and these data were validated by involving experts to account for the need for research from different regional sectors with different professions relevant to the study city region. This methodology constitutes an ES assessment, requiring data of monetary values. The Granger causality test was used to investigate causality between land-use dynamics and ecosystem services variables using time-series data of the last 47 years. The Granger causality test was used to test whether there were positive and negative causal relationships during the study period.

**Table 3.** Accuracy assessment: error matrix.

| Classified Classes | Reference Data | | | | | | | | | |
|---|---|---|---|---|---|---|---|---|---|---|
| | Built-Up Area | Natural Forest | Plantation Forest | Waterbody | Cultivated Land | Grassland | Bushes and Shrubs | Riverside Vegetation | Bare Land | Total Number of Reference Sites |
| Built-up area | 48 | 0 | 0 | 0 | 2 | 0 | 0 | 0 | 0 | 50 |
| Natural forest | 0 | 44 | 3 | 0 | 0 | 0 | 0 | 3 | 0 | 50 |
| Plantation forest | 0 | 0 | 47 | 0 | 0 | 0 | 0 | 3 | 0 | 50 |
| Water bodies | 0 | 0 | 0 | 49 | 0 | 0 | 0 | 1 | 0 | 50 |
| Cultivated land | 0 | 0 | 0 | 0 | 46 | 4 | 0 | 0 | 0 | 50 |
| Grass land | 0 | 0 | 0 | 0 | 3 | 45 | 0 | 2 | 0 | 50 |
| Bushes and shrubs | 0 | 0 | 0 | 0 | 0 | 0 | 48 | 2 | 0 | 50 |
| River side vegetation | 1 | 0 | 5 | 0 | 0 | 4 | 1 | 39 | 0 | 50 |
| Bare land | 2 | 0 | 0 | 0 | 5 | 0 | 0 | 0 | 43 | 50 |
| Column total Totals | 48 | 44 | 47 | 49 | 46 | 45 | 48 | 39 | 43 | 409 |

Source: Field survey, 2019.

**Table 4.** Kappa statistics.

| S. N | Class Name | Kappa Statistics for Each Class |
|:---:|:---:|:---:|
| 1 | Built-up area | 0.9518 |
| 2 | Natural forest | 0.8906 |
| 3 | Plantation forest | 0.9515 |
| 4 | Water bodies | 0.6637 |
| 5 | Cultivated land | 0.9392 |
| 6 | Grassland | 0.9540 |
| 7 | Bushes and shrubs | 0.7743 |
| 8 | Riverside vegetation | 0.4966 |
| 9 | Bare land | 0.7743 |

Source: Field survey, 2019.

*2.3. Data Analysis*

Digital satellite images were processed, classified, and analyzed by the world's most widely used ERDAS Imagine remote sensing software package. The classified data were further analyzed for change detection. Computations or calculations of the area and changes in the land-use categories were made using Arc GIS 10.5.1. This study focuses on the aggregate scale valuation of ecosystem services. The monetary assessment was analyzed using descriptive and using statistical models. The Granger causality test was used to investigate causality between land-use dynamics and ecosystem services variables using time-series data of the last 47 years in Eviews. This indicates both the direction (positive or negative) and the strength of the relationship.

The qualitative data were analyzed using narrations. The climate variability in the study area was analyzed using the software XLSTAT and was employed. This software was also used to generate hypothesis tests of the research questions of the study. This helped to explain the local effects of climate changes on ecosystem services which can help to design mitigation and adaptation actions to the adverse effects of climate change using metrological data. The different policies and spatial plans were analyzed using document review which focuses on the content of communication, as well as interpretive policy analysis that focuses on the meanings of a policy including which values are expressed. For qualitative data interviews and FGD, narration was used to facilitate the analysis of the research.

**3. Results**

*3.1. Land-Use/Land-Cover Change*

The multi-temporal LULC results and the variations for the years 1972, 1984, 1992, 2001, 2012, and 2019 are shown in Figure 2. In 1972, the study area was occupied by different classes of built-up area, natural and plantation forest, waterbody, cultivated land, grassland, bushes and shrubs, riverside vegetation, and bare lands. It is evident from the results that the LULC changes were of the highest amount in agriculture, different vegetations, and water body from 1972 to 2019. Comparison of LULC in 1972 and 2019 derived from satellite imagery interpretation indicates that the built-up area, comprising human habitation developed for nonagricultural uses like building, transport, and communications, is mainly broadened. The agricultural lands which are used for the production of food, vegetables, and other mixed varieties like fruits are decreasing. The study area observed a huge amount of agricultural land converted into settlements and other urban development activities. Water bodies decreased due to the gradual conversion of water spread area into the built-up area. Dense forest comprising all land with a tree cover of canopy density was significantly declined.

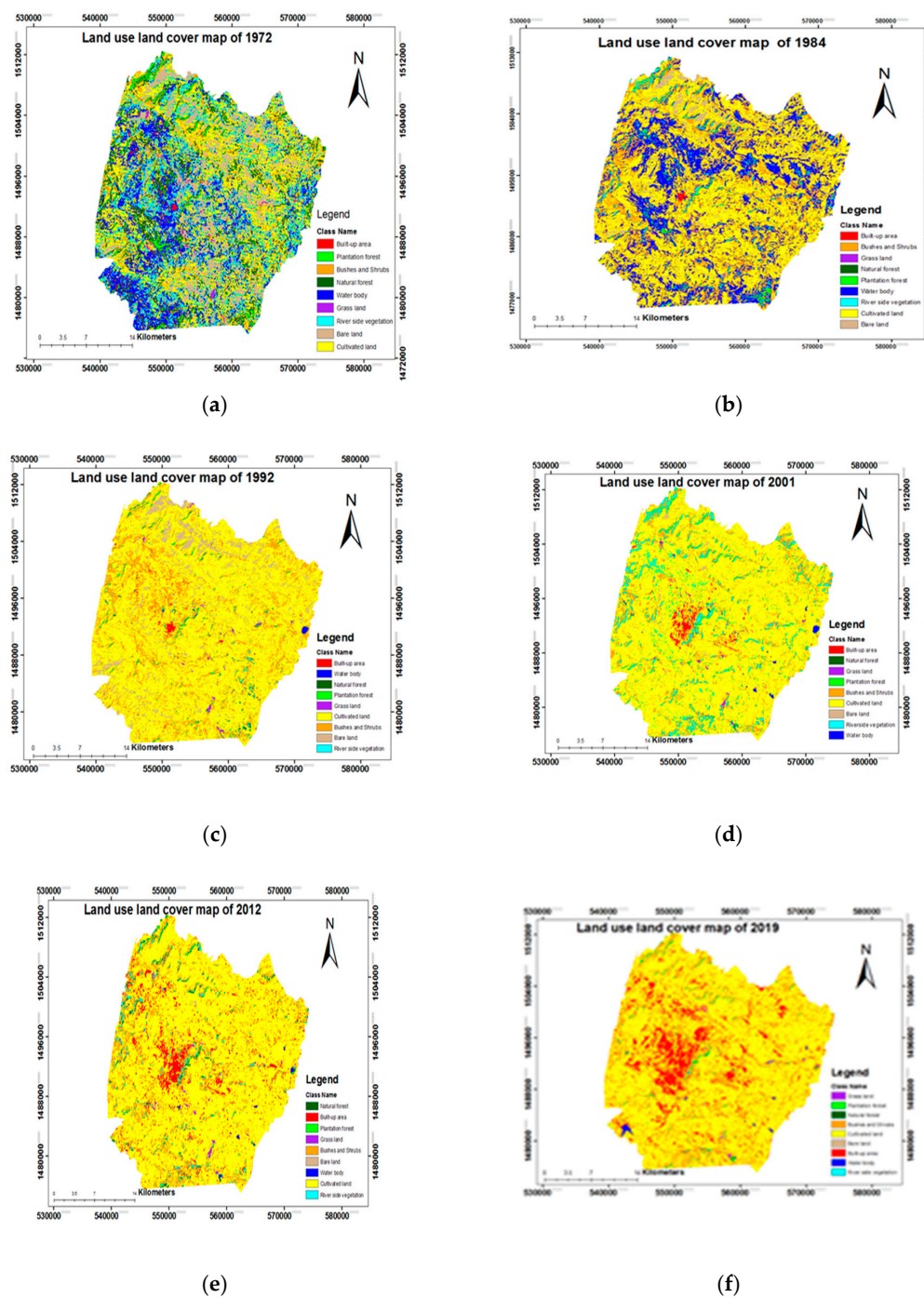

**Figure 2.** Land-use maps of the Mekelle city region in the years 1972 (**a**), 1984 (**b**), 1992 (**c**), 2001 (**d**), 2012 (**e**), and 2019 (**f**).

*3.2. Ecosystem Service Value Changes Results*

The valuation of ecosystem services at the city-region level is confronted here with data scarcity. To offset the weakness, historical LULC data (Table 5) from the period 1972–2019 are the key data used for ecosystem-service values for the study area. Then, the values for ecosystem services per unit area by biome are multiplied by the total area of each biome and the overall ecosystem services and biomes are summed overall. The classified images from the period 1972–2019 are compared with the representative biomes, i.e., LULC types to obtain their corresponding ESV coefficients identified.

**Table 5.** Land-use/land-cover (LULC) types with their area in hectares for the years 1972, 1984,1992, 2001, 2012, and 2019 and biome counterparts with the corresponding value coefficients (USD ha$^{-1}$ year$^{-1}$) modified conservative value coefficients mainly based on the TEEB database [18].

| S. N | Class Name | Area in Hectares (1972–2019) | | | | | | Equivalent Biome | (USD ha$^{-1}$ year$^{-1}$) |
|------|-----------|------|------|------|------|------|------|------|------|
| | | 1972 | 1984 | 1992 | 2001 | 2012 | 2019 | | |
| 1 | Built-up area | 497 | 536 | 541 | 1029 | 4420 | 8897 | Settlement | 0 |
| 2 | Natural forest | 5543 | 1011 | 647 | 181 | 147 | 106 | Tropical forest | 986.69 |
| 3 | Plantation forest | 2082 | 1118 | 529 | 398 | 292 | 497 | Tropical forest | 986.69 |
| 4 | Water body | 15,577 | 14,209 | 2024 | 1207 | 130 | 263 | Lakes/rivers fresh water | 8103.5 |
| 5 | Cultivated land | 26,403 | 40,156 | 65,700 | 70,003 | 69,610 | 64,843 | Cropland | 225.56 |
| 6 | Grass land | 2359 | 1049 | 310 | 297 | 445 | 330 | Grass/rangelands | 293.25 |
| 7 | Bushes and Shrubs | 8183 | 16,149 | 12,559 | 9404 | 10,982 | 13,215 | Wood land | 986.69 |
| 8 | River side vegetation | 11,132 | 9359 | 1277 | 4907 | 1430 | 1235 | Wood land | 986.69 |
| 9 | Bare land | 17,936 | 6134 | 6125 | 2286 | 2256 | 326 | Desert | 0 |
| | Total | 89,712 | 89,712 | 89,712 | 89,712 | 89,712 | 89,712 | | |

To estimate ESVs, locally relevant ecosystem services for this study were identified. To ensure the applicability of the transferred data to the study area, values from tropical areas are applied. The study employed modified conservative value coefficients developed by [18] to contextualize it to the Ethiopian and Mekelle city-region context. Based on the model quantifications of ESV changes induced by LULC using a conservative coefficient, a benefit transfer approach was applied. This value coefficient is assumed to be suitable for the Mekelle city region. We followed the ESVs methodology used by [10,18–21]. To estimate ESV, the following equation is used.

$$ESV = \Sigma \, (A_k \times VC_k)$$

where ESV is the total estimated ecosystem service value, Ak is the area in (ha), and VCk the corresponding value coefficient (USD ha$^{-1}$ yr$^{-1}$) for each LULC category k. The total value of ecosystem services for a particular LULC type for the years 1972, 1984, 1992, 2001, 2012, and 2019 was obtained by the following method: the area of each LULC type in hectare was multiplied($\times$) by its corresponding value coefficients. The values for the LULC types in each reference year were summed to estimate the total ESV of the landscape for each reference year (Table 6).

The Total ESV per year is given as a sum of each value by LULC type.

Using the coefficient values of each LULC category, the ecosystem-service values were estimated in the city region for the years 1972, 1984, 1992, 2001, 2012, and 2019 (Table 2). The total estimated ESVs of the whole study land in millions of USD were about 160.6, 152.1, 46.1, 40.4, 29.5, and 32 million in the years 1972, 1984, 1992, 2001, 2012, and 2019, respectively. The amount of ecosystem service value fluctuated among LULC types of the entire study area across space and time.

At the beginning of the study period, 1972, the natural forest, plantation forest, water body, cultivated land, grass land, bushes and shrubs, river side vegetation accounted in millions USD were 5.5 (3.42%), 2.5 (1.56%), 126.2 (78.59%), 6 (3.74%), 0.691 (0.42%), 8.7 (5.42%) and 11 (6.85%), respectively. In 1984, natural forest, plantation forest, water body, cultivated land, grass land, bushes and shrubs, river side vegetation accounted in millions USD for 1 (0.66%), 1.1 (0.72%), 115.1 (75.62%), 9.58 (6.3%), 0.307 (0.2%), 15.9 (10.5%) and 9.2 (6%), respectively. In 1992, natural forest, plantation forest, water body, cultivated land, grass land, bushes and shrubs, river side vegetation accounted in millions USD for 0.64 (1.39%), 0.522 (1.12%), 16.4 (35.6%), 14.8 (32.1%), 0.091 (0.19%), 12.4 (26.9%) and 1.26 (2.71%), respectively.

**Table 6.** The ecosystem-service values per hectare of different land-use types in the Mekelle city region.

| S. N | Class Name | 1972 | | 1984 | | 1992 | | 2001 | | 2012 | | 2019 | |
|---|---|---|---|---|---|---|---|---|---|---|---|---|---|
| | | ESV | % | ESV | % | ESV | % | ESV | % | ESV | % | ESV | % |
| 1 | Built-up area | 0.0 | 0.0 | 0.0 | 0.0 | 0.0 | 0.0 | 0.0 | 0.0 | 0.0 | 0.0 | 0.0 | 0.0 |
| 2 | Natural forest | 5.5 | 3.42 | 1 | 0.66 | 0.64 | 1.39 | 0.18 | 0.45 | 0.145 | 0.49 | 0.105 | 0.33 |
| 3 | Plantation forest | 2.5 | 1.56 | 1.1 | 0.72 | 0.522 | 1.12 | 0.393 | 0.98 | 0.288 | 0.98 | 0.490 | 1.53 |
| 4 | Water body | 126.2 | 78.59 | 115.1 | 75.62 | 16.4 | 35.6 | 9.8 | 24.3 | 1 | 3.54 | 2.1 | 6.56 |
| 5 | Cultivated land | 6 | 3.74 | 9.58 | 6.3 | 14.8 | 32.1 | 15.8 | 39.1 | 15.7 | 53.2 | 14.6 | 45.6 |
| 6 | Grass land | 0.691 | 0.42 | 0.307 | 0.2 | 0.091 | 0.19 | 0.087 | 0.27 | 0.130 | 0.44 | 0.097 | 0.30 |
| 7 | Bushes and Shrubs | 8.7 | 5.42 | 15.9 | 10.5 | 12.4 | 26.9 | 9.3 | 23 | 10.8 | 36.6 | 13.4 | 41.9 |
| 8 | River side vegetation | 11 | 6.85 | 9.2 | 6 | 1.26 | 2.71 | 4.8 | 11.9 | 1.4 | 4.75 | 1.21 | 3.78 |
| 9 | Bare land | 0.0 | 0.0 | 0.0 | 0.0 | 0.0 | 0.0 | 0.0 | 0.0 | 0.0 | 0.0 | 0.0 | 0.0 |
| | Total | 160.6 | 100 | 152.1 | 100 | 46.1 | 100 | 40.4 | 100 | 29.5 | 100 | 32 | 100 |

In 2001, natural forest, plantation forest, waterbody, cultivated land, grassland, bushes and shrubs, riverside vegetation accounted in millions USD were 0.18 (0.45%), 0.393 (0.98%), 9.8 (24.3%), 15.8 (39.1%), 0.087 (0.27%), 9.3 (23%) and 4.8 (11.9%), respectively. In 2012, natural forest, plantation forest, water body, cultivated land, grass land, bushes and shrubs, river side vegetation accounted for, in millions USD, 0.145 (0.49%), 0.288 (0.98%), 1 (3.54%), 15.7 (53.2%), 0.130 (0.44%), 10.8 (36.6%) and 1.4 (4.75%), respectively. In 2019, natural forest, plantation forest, water body, cultivated land, grass land, bushes and shrubs, river side vegetation accounted for, in millions USD, 0.105 (0.33%), 0.490 (1.53%), 2.1 (6.56%), 14.6 (45.6%), 0.097 (0.30%), 13.4 (41.9%) and 1.2 (3.78%), respectively. The change in ecosystem service value is estimated by calculating the differences between the estimated values for each LULC category of the selected years. The percentage changes in ESV between each year are calculated based on:

$$\% \text{ Changes in ESV} = (ESV_{t2} - ESV_{t1}) \, / \, ESV_{t1} * 100$$

where $ESV_{t2}$ (USD ha$^{-1}$ yr$^{-1}$) is the total ecosystem service value of the final year, and ESV (USD ha$^{-1}$ yr$^{-1}$) is the ecosystem service value of the initial year.

### 3.2.1. Variations in Ecosystem-Service Values between the Study Periods

Table 7 demonstrates that the change in ESVs of each LULC type also showed a considerable reduction in ESVs. The value of forest continued to decline throughout the study period of 1972–1984, 1984–1992, 1992–2001, 2001–2012, 2012–2019 in USD million by 4.5 million, 0.36 million, 0.46 million, 0.27 million, and 0.035 million. The natural forest ESVs were reduced by 5.39 million (−3793%) during the whole study period of 1972–2019. Similarly, plantation declined in the study periods by 1.4 million, 0.578 million, 0.129 million, 0.105 million, and 0.202 million consecutively. The plantation forest ESVs were reduced by 2.01 million (−510%) during the whole study period of 1972–2019. Water ESVs rapidly declined by 11.1 million, 98.7 million, 6.6 million, 7.8 million, and 1.1 million consecutively. The water body ESVs reduced by 124.1 million (−6009.5%) during the whole study period of 1972–2019. Conversely, cultivated land showed an increase of 3.58 million, 5.22 million, 1 million and a decrease of 0.1 and 1.1 million and in the last period, it increased by 8.6 million. The cultivated land ESVs increased by 8.6 million 243.3%) during the whole study period of 1972–2019.

**Table 7.** Net changes in ecosystem-service values (ESVs) in each study period.

| S. N | Class Name | Net Changes in ESVS in USD Million per Year | | | | | Changes During 1972–2019 | |
|---|---|---|---|---|---|---|---|---|
| | | 1972–1984 | 1984–1992 | 1992–2001 | 2001–2012 | 2012–2019 | ESVs | % |
| 1 | Built-up area | - | - | - | - | - | - | - |
| 2 | Natural forest | −4.5 | −0.36 | −0.46 | −0.27 | −0.035 | −5.39 | −3793 |
| 3 | Plantation forest | −1.4 | −0.578 | −0.129 | −0.105 | −0.202 | −2.01 | −510 |
| 4 | Water body | −11.1 | −98.7 | −6.6 | −7.8 | 1.1 | −124.1 | −6009.5 |
| 5 | Cultivated land | 3.58 | 5.22 | 1 | −0.1 | −1.1 | 8.6 | 243.3 |
| 6 | Grass land | −0.384 | −0.216 | −0.04 | 0.043 | −0.033 | −0.594 | −712 |
| 7 | Bushes and Shrubs | 7.2 | -3.5 | −3.1 | 1.5 | 2.6 | 4.7 | 64.9 |
| 8 | River side vegetation | −1.8 | −7.94 | 3.54 | −3.4 | −0.19 | −9.79 | −909 |
| 9 | Bare land | - | - | - | - | - | | |

Positive ESVs indicate an increase whereas negative ESVs imply a decrease in amount.

The grassland continued to decline throughout the study period of 1972–1984, 1984–1992, 1992–2001, 2001–2012, 2012–2019 by 0.384 million, 0.216 million, 0.04 million and increased by 0.043 million and again declined by 0.033 million, respectively. In sum, during the whole study period, it decreased by 0.594 (−712%). Bushes and Shrubs in the starting period of 1972–1984 increased by 7.2 million, and in 1984–1992 and 1992–2001, declined by 3.5 and 3.1 million. However, from 2001–2012 and 2012–2019, they increased by 1.5 and 2.6 million. The ESVs of the classes showed an increment of 4.7 million (64.9%) during the whole study period of 1972–2019. Despite slight increases in riverside vegetation in 1992-2001 by 3.54 million, it showed a continuous decline. It decreased in 1972–1984, 1984–1992, 2001–2012, 2012–2019 in USD million by 1.8 million, 7.94 million, 3.4 million, and 0.19 million consequently. This LULC class type decreased the ESVs by 9.79 million (−909%) during the whole study period.

During the study period, ESV variations revealed a significant reduction in the total ecosystem-service values (Figure 3). Over the whole study period from 1972–2019, a loss of USD 128.6 million was observed which is a reduction of 501.9% (Figure 2). The total amount of changes of ESVs during the 1972–1984 period was about USD 8.5 million, which is about 5.29% of the value reduction that existed in 1972. The total ESV was more decreased between 1984–1992 with an amount of about USD 106 million which is a decline of 329.9% in the value that existed in 1984. This reduction in ESVs was severe due to climate changes and drought. In the 1992–2001 period, it decreased by USD 5.7 million which is a decline of 12.36% in the value that existed in 1992. The next period between 2001–2012 also decreased by USD 10.9 million which is a reduction of 26.98% of the year 2001. However, in the last period between 2012–2019, it showed an increment from the previous study period by 2.5 USD which is an increase of 7.81%. This increment during the period 2012–2019 showed an ecological restoration due to IWSM and other policy interventions.

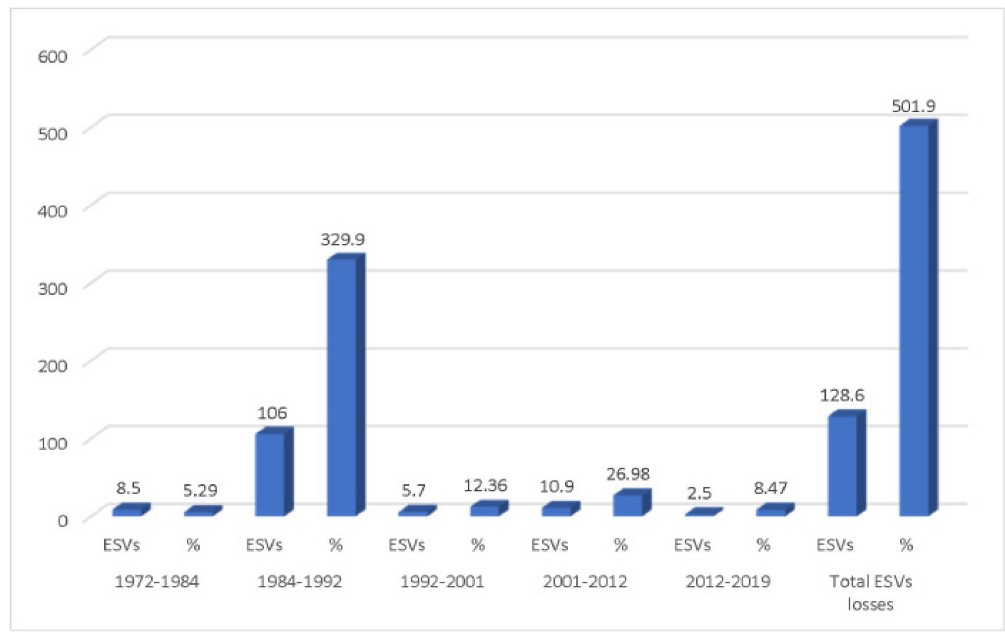

**Figure 3.** Total ecosystem-service value (ESV) loss during the period 1972–2019 (in USD million).

### 3.2.2. Estimated Services of Individual Ecosystem Functions and Their Changes

The effects of land-use dynamics on individual ecosystem-service values were estimated. The individual ecosystem service values were multiplied with each ESV identified for each LULC class because there is a direct relationship between each ESV and the LULC type. The values of individual ecosystem services function provided by each LULC category at the city-region level were calculated by the following equation.

$$ESVf = \Sigma \, (Ak \times VCfk)$$

where $ESV_f$ is the estimated ecosystem service value of function f, $A_k$ is the area (ha), and $VC_{fk}$ the value coefficient of function f (USD ha$^{-1}$ yr$^{-1}$) for each LULC category k. The contributions of each ecosystem function to the overall value of ecosystem services per year were hierarchical, based on an estimated value of ecosystem functions for each reference year and summarized in Table 8.

The model adopted in the study classified ecosystem service into 17 types of ES (Table 9). The equivalent weight factor of ecosystem-service values per hectare in Ethiopia was applied.

**Table 8.** Annual value coefficients for ecosystem service functions of each land-use/land-cover (LULC) type.

| Ecosystem Services | | Each LULC Types Ecosystem-Service Values (USD/ha/yr) | | | | | | | | | |
|---|---|---|---|---|---|---|---|---|---|---|---|
| | | Built-Up Area | Natural Forest | Plantation Forest | Water Body | Cultivated Land | Grass Land | Bushes and Shrubs | Riverside Vegetation | Bare Land | Total ESV |
| Provisioning Services | Water supply | 0 | 8 | 8 | 2117 | 0 | 0 | 8 | 8 | 0 | 2149 |
| | Food production | 0 | 32 | 32 | 41 | 187.56 | 117.45 | 32 | 32 | 0 | 474.01 |
| | Raw material | 0 | 51.24 | 51.24 | 0 | 0 | 0 | 51.24 | 51.24 | 0 | 204.96 |
| | Genetic resources | 0 | 41 | 41 | 0 | 0 | 0 | 41 | 41 | 0 | 164 |
| Regulating Services | Water regulation | 0 | 6 | 6 | 5445 | 0 | 3 | 6 | 6 | 0 | 5472 |
| | Water treatment | 0 | 136 | 136 | 431.5 | 0 | 87 | 136 | 136 | 0 | 1062.5 |
| | Erosion control | 0 | 245 | 245 | 0 | 0 | 29 | 245 | 245 | 0 | 1009 |
| | Climatic regulation | 0 | 223 | 223 | 0 | 0 | 0 | 223 | 223 | 0 | 892 |
| | Biological control | 0 | 0 | 0 | 0 | 24 | 23 | 0 | 0 | 0 | 47 |
| | Gas regulation | 0 | 13.68 | 13.68 | 0 | 0 | 7 | 13.68 | 13.68 | 0 | 61.72 |
| | Disturbance regulation | 0 | 5 | 5 | 0 | 0 | 0 | 5 | 5 | 0 | 20 |
| Supporting services | Nutrient cycling | 0 | 184.4 | 184.4 | 0 | 0 | 0 | 184.4 | 184.4 | 0 | 737.6 |
| | Pollination | 0 | 7.27 | 7.27 | 0 | 14 | 25 | 7.27 | 7.27 | 0 | 68.08 |
| | Soil formation | 0 | 10 | 10 | 0 | 0 | 1 | 10 | 10 | 0 | 41 |
| | Habitat | 0 | 17.3 | 17.3 | 0 | 0 | 0 | 17.3 | 17.3 | 0 | 69.2 |
| Cultural services | Recreation | 0 | 4.8 | 4.8 | 69 | 0 | 0.8 | 4.8 | 4.8 | 0 | 89 |
| | Cultural | 0 | 2 | 2 | 0 | 0 | 0 | 2 | 2 | 0 | 8 |
| | Total ESV | 0 | 986.69 | 986.69 | 8103.5 | 225.56 | 293.25 | 986.69 | 986.69 | 0 | 12,569.07 |

**Table 9.** The annual estimated value of ecosystem functions ($ESV_f$ in USD million USD year$^{-1}$) under each service category from 1972 to 2019.

| Ecosystem Services | | The Annual Estimated Value of Ecosystem Functions ($ESV_f$ in USD Million USD Year$^{-1}$) | | | | | | |
|---|---|---|---|---|---|---|---|---|
| | | $ESV_{f1972}$ | $ESV_{f1984}$ | $ESV_{f1992}$ | $ESV_{f2001}$ | $ESV_{f2012}$ | $ESV_{f2019}$ | Over All Changes |
| Provisioning Services | Water supply | 27.5 | 26 | 7.8 | 6.9 | 5 | 5.5 | −22 |
| | Food production | 6.1 | 5.7 | 1.8 | 1.5 | 1.1 | 1.2 | −4.9 |
| | Raw material | 2.6 | 2.4 | 0.8 | 0.7 | 0.48 | 0.5 | −2.1 |
| | Genetic resources | 2.1 | 1.9 | 0.6 | 0.5 | 0.38 | 0.4 | −1.7 |
| Regulating Services | Water regulation | 70 | 66.2 | 20 | 17. | 12.9 | 14 | −56 |
| | Water treatment | 13.5 | 12.8 | 3.8 | 3.4 | 2.5 | 2.6 | −10.9 |
| | Erosion control | 12.8 | 12.2 | 3.6 | 3.2 | 2.4 | 2.6 | −10.2 |
| | Climatic regulation | 11.3 | 10.7 | 3.3 | 2.9 | 2.1 | 2.3 | −9 |
| | Biological control | 0.6 | 0.5 | 0.2 | 0.15 | 0.11 | 0.12 | −0.48 |
| | Gas regulation | 0.8 | 0.7 | 0.2 | 0.19 | 0.14 | 0.16 | −0.64 |
| | Disturbance regulation | 0.3 | 0.2 | 0.1 | 0.06 | 0.05 | 0.06 | −0.24 |
| Supporting services | Nutrient cycling | 9.5 | 8.9 | 2.7 | 2.4 | 1.75 | 1.9 | −7.6 |
| | Pollination | 0.9 | 0.8 | 0.2 | 0.21 | 0.002 | 0.17 | −0.73 |
| | Soil formation | 0.5 | 0.4 | 0.2 | 0.13 | 0.096 | 0.1 | −0.4 |
| | Habitat | 0.9 | 0.8 | 0.3 | 0.22 | 0.162 | 0.18 | 0.72 |
| Cultural services | Recreation | 1.1 | 1 | 0.3 | 0.28 | 0.21 | 0.22 | −0.88 |
| | Cultural | 0.1 | 0.96 | 0.29 | 0.06 | 0.02 | 0.03 | −0.07 |
| | Total | 160.6 | 152.1 | 46.1 | 40.4 | 29.5 | 32 | −128.6 |

Positive ESVs indicate an increase whereas negative ESVs imply a decrease in ESV function amount.

From the above, 17 individual ecosystem services based on their contributions from high to low to the overall ecosystem-service values in millions mainly came from the provisioning category water supply and food production and regulating water regulation and water treatment, from supporting services nutrient cycling and pollination, and from the cultural categories of both recreation and culture, respectively. This order of contribution to the various services changed over the different study periods. The aggregate contribution was about USD 160.6, 152.1, 46.1, 40.4, 29.5, and 32 in the years 1972, 1984, 1992, 2001, 2012, and 2019, respectively. When comparing the values concerning service categories of the year 1972, the results revealed the highest value for the group of regulating services, i.e., water regulation was USD 70 million, then provisioning water supply with 27.5 million, followed by water treatment at 13.5 million. This order of contribution by service categories remained the same over the different study periods until 2019, with the exception of erosion control. However, the contribution of individual ecosystem services declined throughout the study period (Figure 4).

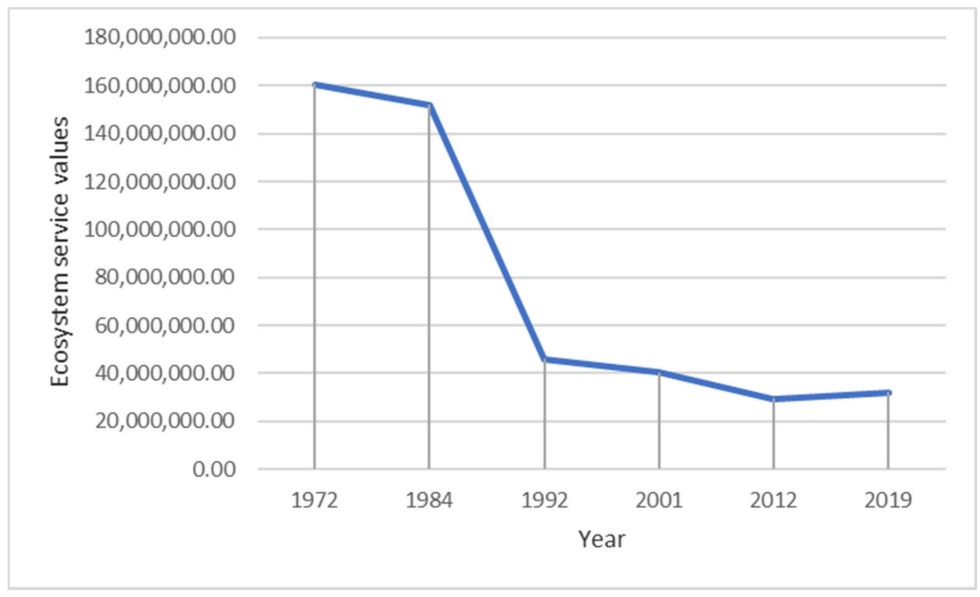

**Figure 4.** Trends of ecosystem service value decline in the Mekelle city region from 1972 to 2019.

The above graph (Figure 3) shows the drastic decline in ecosystem service values in the last five decades. The results clearly show that LULC changes were significant during the period from 1972 to 2019. The study area experienced an enormous spatial change in LULC during the study period. During the period between 1972 and 2019, 60,714.83 ha (67.68%) of the total landscape of the study area was converted from one LULC type to the other. Ecosystem services were strongly influenced by LULC changes. There is a significant expansion in the built-up area. On the other hand, there is a decrease in the agricultural area, water bodies and forest areas. The ES comprises both supplies with the potential to benefit people and demand, i.e., the desired amount of consumption of that supply, which depends on people's desire for and access to ESs. In the city region, the benefits to the city residents arise from the interaction of supply and demand of watershed ecosystem services. Due to the continuing rise in population accompanied by rapid urban expansion and other factors such as climate changes, LULC changes occur.

As a result of the linear relationship between ecosystem service supply and LULC changes, there was a direct relationship. Beyond that moment, only slight changes were detected, which suggests hope for environmental rehabilitation. Nevertheless, there was a marked reduction in the potential supply of water ES. The landscapes continue to be converted and used in unsustainable ways. Anthropogenic activities and city region land development has triggered intense interference to ecosystems. The changes of ecosystem

service supply were determined more by land-use dynamics, such as the intensification of agricultural production. ES supply was affected by drivers such as temperature, precipitation change, and management practices. The recent slight increase in the provision of ES is noteworthy, especially in the context of the ongoing, larger landscape transformations.

### 3.3. Spatial and Temporal Distribution of Ecosystem-Service Values

Maps of ecosystem service values in the last five decades exhibited the spatial and temporal distribution of ESVs among the nine different types of LULC types (Figure 5). In 1972, the highest ESVs were found in a different part of the study area. The current urban fabric had high ESVs; the southwestern part of the Mekelle city region, western, northern, and eastern parts were rich in ESVS. The lowest ESVs were concentered in some parts of the eastern lowland of the study area. This is due to perennial water bodies which strongly distributed in the riverside vegetation. In 1984, the northern part had the highest ESVs followed by the southwestern part.

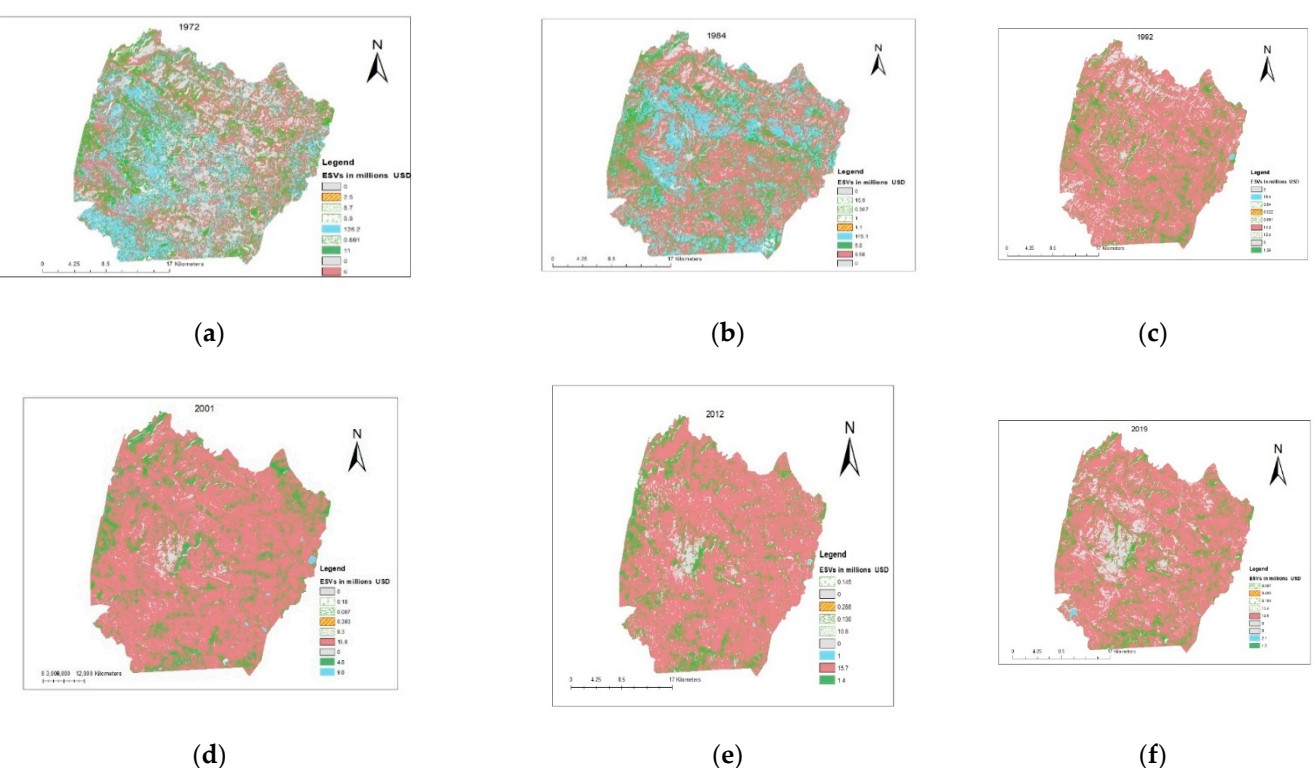

**Figure 5.** The spatial location of ESVs in millions USD for the years 1972 (**a**), 1984 (**b**), 1992 (**c**), 2001 (**d**), 2012 (**e**), 2019 (**f**), Source: Prepared by the author, 2021.

In 1992, ESVs in all parts of the study area dramatically declined. A small amount of ESVs remained. The highest value was derived only from cultivated land. Most of the LULC changes were converted into subsistence farming which had a low contribution to total ESV. In 2001, the ESVs started to recover. Specifically, the riverside vegetation was flourishing again in all parts of the study area. From 2012 onwards, the highest ecosystem service value started to be locally specific. The southwestern part and southern part had the highest ESVs.

In 2019, the highest value was in areas with high coverages of bushes and shrubs. In particular, the northern, western part, and eastern lowlands had rich ESVs. The lowest ESVs mostly occurred in the sleepy slopes. Currently, specific areas with the highest ecosystem service values are found around the riversides. Specifically, this pertains to the southwestern part of the study area around the Geba dam and the southern part near the Gereb Segen dam. The core city of Mekelle has low ecosystem-service values due to

built-up area expansion. The northern part and eastern part are dominated by bushes and shrubs, along with plantation forests.

### 3.4. Granger Causality Test between Land-Use Dynamics and ESVs

The data of ESVs and LULC changes were used to analyze the relationship between ecosystem-service values and land-use dynamics. In the Mekelle city region, there exists ESV variation due to LULC changes. Therefore, statistical analyses were employed to evaluate how the LULC changes affect multiple ESVs. Before the Granger causality test was made, the study tested for stationarity in the variables using the augmented Dickey–Fuller (ADF) procedure in Eviews (Table 10).

**Table 10.** Augmented Dickey–Fuller test statistic.

| **Null Hypothesis: Ecosystem Service Value has a Unit Root** | | | | |
|---|---|---|---|---|
| **Exogenous: Constant** | | | | |
| **Lag Length: 0 (Automatic-Based on SIC, maxlag = 9)** | | | | |
| | | | t-Statistic | Prob.* |
| Augmented Dickey–Fuller test statistic | | | −6.711728 | 0.0000 |
| Test critical values: | 1% level | | −3.600987 | |
| | 5% level | | −2.935001 | |
| | 10% level | | −2.605836 | |
| | Augmented Dickey–Fuller Test Equation | | | |
| | Dependent Variable: D(ECOSYSTEM_SERVICE_VALUE) | | | |
| | Method: Least Squares | | | |
| | Date: 10/02/21 Time: 16:37 | | | |
| | Sample (adjusted): 2 42 | | | |
| | Included observations: 41 after adjustments | | | |
| Variable | Coefficient | Std. Error | t-Statistic | Prob. |
| ECOSYSTEM_SERVICE_VALUE(-1) | −1.073208 | 0.159900 | −6.711728 | 0.0000 |
| C | 11.92331 | 4.448895 | 2.680060 | 0.0107 |
| R-squared | 0.535976 | Mean dependent var | | −0.104634 |
| Adjusted R-squared | 0.524077 | S.D. dependent var | | 37.79469 |
| S.E. of regression | 26.07348 | Akaike info criterion | | 9.407265 |
| Sum squared resid | 26513.22 | Schwarz criterion | | 9.490854 |
| Log-likelihood | −190.8489 | Hannan–Quinn criterion. | | 9.437703 |
| F-statistic | 45.04729 | Durbin–Watson stat | | 2.015051 |
| Prob (F-statistic) | 0.000000 | | | |

\* MacKinnon (1996) one-sided $p$-values. \* denotes rejection of the hypothesis at the 0.05 level. The unit root tests using the Augmented Dickey–Fuller test reveals that all the variables are non-stationary. Johansen's co-integration test (Table 11) discloses that at a 5% level of significance there is at least two co-integrating equations in the study.

The test reveals that there are at least two co-integrated series out of 42 observations considered in this study.

The ecosystem-service values were directly related to ecosystem types, i.e., LULC causes changes in ES. The unprecedented LULC changes were a result of natural and anthropogenic activities, resulting in adverse impacts on ecosystems and affecting their values. To quantitively understand the relationship between ESVs and land-use dynamics, the Granger causality test was conducted (Table 12). It cannot reject the hypothesis that ecosystem-service values do not Granger-cause land-use dynamics, but this study does reject the hypothesis that land-use dynamics do not Granger-cause ecosystem service values. Hence, it appears that Granger causality runs one-way from land-use dynamics to ecosystem-service values and not the other way.

**Table 11.** Johansen co-integration test.

Date: 10/02/21 Time: 16:54
Sample (adjusted): 3 42
Included observations: 40 after adjustments
Trend assumption: Linear deterministic trend
Series: ECOSYSTEM_SERVICE_VALUE LAND_USE_DYNAMICS
Exogenous series: LAND_USE_DYNAMICS
Lags interval (in first differences): 1 to 1
Unrestricted Cointegration Rank Test (Trace)

| Hypothesized No. of CE(s) | Eigenvalue | Trace Statistic | 0.05 Critical Value | Prob.** |
|---|---|---|---|---|
| None | 1.000000 | NA | 15.49471 | NA |
| At most 1 * | 0.368069 | 18.35899 | 3.841465 | 0.0000 |

Trace test indicates two cointegrating equation(s) at the 0.05 level
Unrestricted Cointegration Rank Test (Maximum Eigenvalue)

| Hypothesized No. of CE(s) | Eigenvalue | Max-Eigen Statistic | 0.05 Critical Value | Prob.** |
|---|---|---|---|---|
| None | 1.000000 | NA | 14.26460 | NA |
| At most 1 * | 0.368069 | 18.35899 | 3.841465 | 0.0000 |

Max-eigenvalue test indicates 2 cointegrating eqn(s) at the 0.05 level
Unrestricted Cointegrating Coefficients (normalized by b'*S11*b = I):

| ECOSYSTEM_SERVICE_VALUE | LAND_USE_DYNAMICS |
|---|---|
| $1.39 \times 10^{-17}$ | $8.39 \times 10^{-5}$ |
| 0.060851 | $-2.42 \times 10^{-5}$ |

Unrestricted Adjustment Coefficients (alpha):

| D(ECOSYSTEM_SERVICE_VALUE) | $-8.487530$ | $-18.90457$ |
|---|---|---|
| D(LAND_USE_DYNAMICS) | $-11925.19$ | $9.09 \times 10^{-13}$ |

| 1 Cointegrating Equation(s): | Log-likelihood | NA |
|---|---|---|

Normalized cointegrating coefficients (standard error in parentheses)

| ECOSYSTEM_SERVICE_VALUE | LAND_USE_DYNAMICS |
|---|---|
| 1.000000 | $9.03 \times 10^{-12}$ |
| | (NA) |

Adjustment coefficients (standard error in parentheses)

| D(ECOSYSTEM_SERVICE_VALUE) | $-7.88 \times 10^{-12}$ |
|---|---|
| | $(4.9 \times 10^{-17})$ |
| D(LAND_USE_DYNAMICS) | $-1.11 \times 10^{-13}$ |
| | (NA) |

* denotes rejection of the hypothesis at the 0.05 level; ** MacKinnon–Haug–Michelis (1999) *p*-values.

**Table 12.** Pairwise Granger Causality Tests.

Date: 10/02/21 Time: 15:26
Sample: 1 42
Lags: 2

| Null Hypothesis: | Obs | F-Statistic | Prob. |
|---|---|---|---|
| ECOSYSTEM_SERVICE_VALUE does not Granger Cause LAND_USE_DYNAMICS | 31 | 9.56449 | 0.0018 |
| LAND_USE_DYNAMICS does not Granger Cause ECOSYSTEM_SERVICE_VALUE | | 10.6264 | 0.0013 |

At Alpha ($\alpha$) = 0.05. Decision rule: reject $H_0$ if *p*-value < 0.05.

It is inferred that there was a relationship between land-use change and ecosystem service value. Since the *p*-value of land-use dynamics, which does not Granger-cause ecosystem-service values, is 0.0013 which is <0.05, the hypothesis ($H_o$) that temporal and spatial land-use/land-cover dynamics do not affect ecosystem service value is rejected.

*3.5. Qualitative Data Analysis*

In the FGD, the participants were selected purposely for experts who have knowledge and experience in the ecosystem, and discussions were held with 12 experts. The discussion was prepared based on a checklist. The participants of FGD carefully recruited six experts per group. The environment was comfortable, circle seating was recorded by using tape

and predetermined questions, and open-ended questions were prepared for discussion. The author was a moderator and an assistant moderator was selected to handle notes.

The qualitative results here are used to increase the understanding of ESVS among experts. The results of focus group discussions with experts on ESVs of the study area show that there is limited awareness of ES valuations. The main difference between the quantitative ESVs and focus groups was in the understanding of not only our study area but also on consensus, conflicting perceptions, and lack of information on ESVs. Based on the results, the study developed some first remarks which became guiding essentials for the later stages of the study. Among the FGD discussants, there was a common understanding that ESVs have great potential to sustain ecosystem services and for policy implications, but those were largely not yet developed. Major ecosystem service categories were highlighted by the experts' provisioning, regulating, and cultural and supporting services. The qualitative analysis revealed the importance of the ESV method employed in the study.

The experts helped in identifying the most relevant services, their values and trade-offs, and the main policy objectives. The discussion revealed that the experts had very different interests related to the ecosystem. They also tend to attach more ESVs to provisioning services (crops, raw materials, medicinal resources), regulating (carbon sequestration, climate regulation, purification of water) and cultural (recreation, spiritual enrichment, cognitive development, aesthetic) services. The FGD participants had a common understanding that the Mekelle city region watersheds have great potential to provide bundles of ecosystem services. The results of this study demonstrate the value of combining qualitative and quantitative methods to improve the reliability and validity of ecosystem-service values. FGDs are demonstrated to be paramount to understanding the attitudes towards valuation concepts and their importance for environmental sustainability. The FGDs prove to be paramount to understanding the underlying attitudes and motives towards the proposed scenarios and their institutional context. FGDs provide the possibility to identify the specific terms and conditions on which respondents would accept land-use change scenarios and help to understand preferences regarding the distribution of costs and benefits over time. The study found that FGDs are very beneficial to support ESV quantifications.

## 4. Discussion

### 4.1. Factors Driving LULCC

4.1.1. Climate Variability

The study scrutinizes the historical climatic from 1972 to 2018 and the consequence this had on the ecosystem services in the Mekelle city region. Data were used from two metrological stations nearby stations in the Mekelle area (Mekelle airport and Mekelle observatory metrological head office). The zero values (Figure 6) are missing values. Some data across the observation were not acquired from metrological agencies.

The Mann–Kendall trend test (Table 13) was used here to detect a trend in a series of values using the XLSTAT statistical software. This is used to identify the trend in a series of the variable of interest of average maximum temperature and monthly total rainfall. The study takes into account the seasonality of the series, i.e., monthly data with a seasonality of 12 months. The Mann–Kendall test was used here to assess statistically whether there was a monotonic decreasing or increasing tendency of the variable of importance over time. A monotonic positive or negative incline indicates that the variable is constantly increasing or decreasing through time; however, the trend may or may not be linear [22]. The trend test on a time series shows the following results.

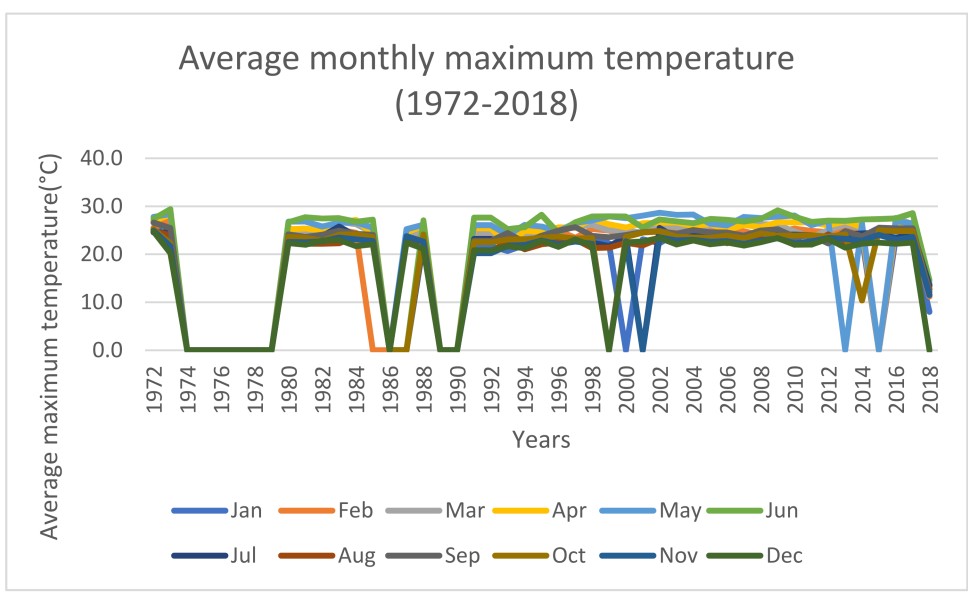

**Figure 6.** Average monthly maximum temperature (1972–2018).

**Table 13.** Mann–Kendall trend test.

| Kendall's tau | 0.278 |
| --- | --- |
| S′ | 10.000 |
| *p*-value (Two-tailed) | 0.001 |
| alpha | 0.05 |

The magnitude of that trend (sen slope) is 7.25. A very high positive value of S is an indicator of an increasing climate variability trend (Table 14). Since the *p*-value is less than 0.05, the study rejected the null hypothesis, i.e., H0: There is no trend in the series. Therefore, there is a trend in the average maximum temperature series. The monthly total rainfall is presented in Figure 7.

**Table 14.** Seasonal Mann–Kendall Test.

| Seasonal Mann–Kendall Test/Period = 12/Serial Dependence/Two-Tailed Test (Year): | |
| --- | --- |
| Kendall's tau | −0.111 |
| S′ | −4.000 |
| *p*-value (Two-tailed) | 0.002 |
| alpha | 0.05 |
| The *p*-value is computed using an exact method. | |
| interpretation: | |
| H0: There is no trend in the series; | |
| Ha: There is a trend in the series | |
| As the computed *p*-value is less than the significance level alpha = 0.05. Therefore, the data show there is a trend. | |
| Sen's slope (Period = 12): | −1.5 |

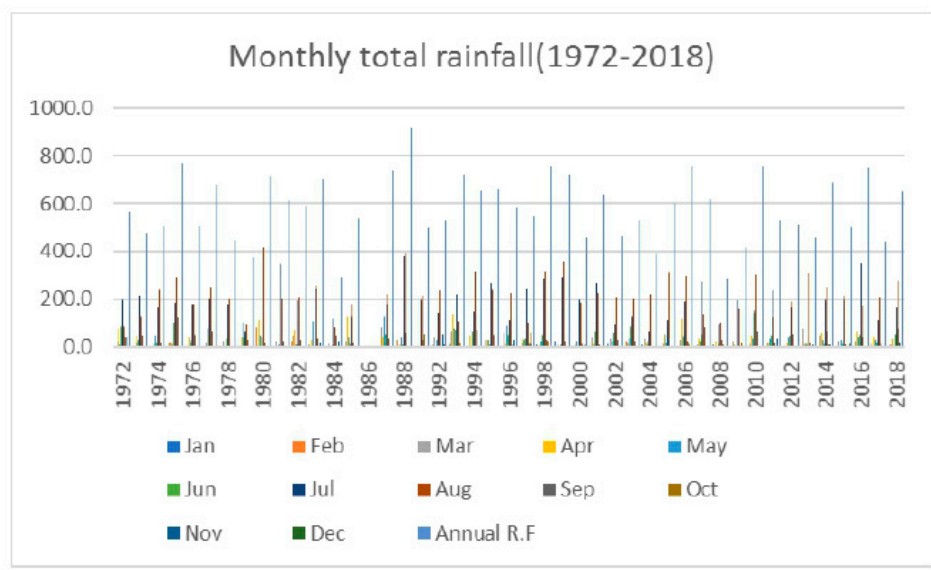

**Figure 7.** Monthly total rainfall (1972–2018).

The overall gain and loss in the ecosystem from the changes for LULC of two time periods of 1984–1992 indicate that USD 106 million declines of ecosystem-service values were registered. This was corroborated by the document review. The document analysis shows that within the period 1984–1992, severe destruction of the ecosystem was observed. Various ecosystems were dramatically reduced. This was further aggravated by mismanagement of natural resources and severe damage to the ecosystem (water bodies, forests, bushes and shrubs, and riverside vegetation). Drought and environmental deterioration have historically imposed heavy costs in Tigrai, Ethiopia, and, in particular, the study area [23]. In 1985, there was a heavy drought. Annual rainfall variation (Figure 8) was observed in the study area. Environmental degradation is severe in the Mekelle city region which is threatening many parts of the study area [14] The disastrous climatic conditions and civil unrest (war between the central government and political party known as TPLF) had significantly contributed to the huge drop in ecosystem-service values. A study by [24] showed decreasing trends in the annual rainy days (1961–2003).

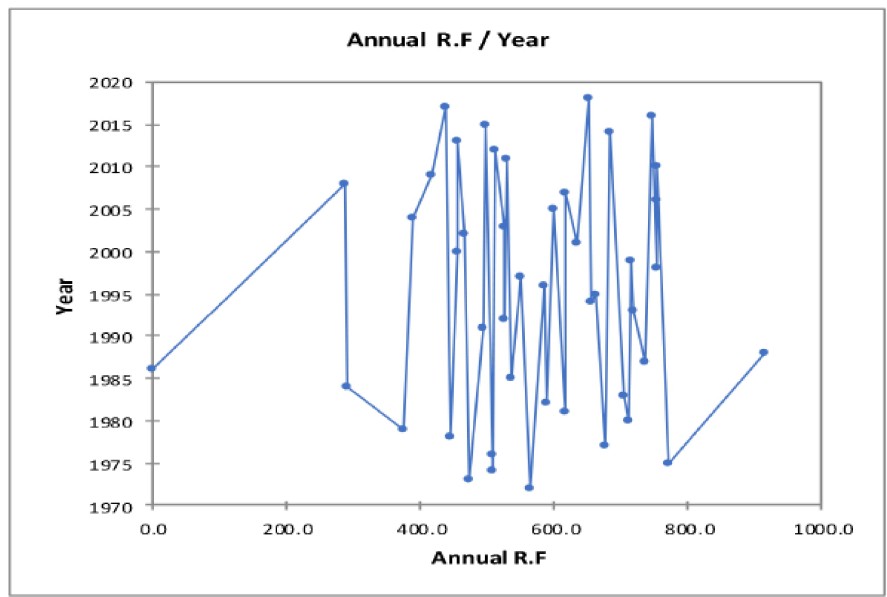

**Figure 8.** Annual rainfall R. F/.

### 4.1.2. Demographic Drivers

The study also identified the continuing increase of population that brought substantial changes to LULC and ecosystems (Table 15).

**Table 15.** The Population of Mekelle city region (1972–2019).

| S. N | Census Year | Population | Population Density |
|------|-------------|------------|--------------------|
| 1 | 1972 | 93,163 | 104 |
| 2 | 1984 | 135,416 | 151 |
| 3 | 1994 | 202,752 | 226 |
| 4 | 2007 | 329,825 | 368 |
| 5 | 2019 | 556,127 | 620 |

Source: (TCSA, 2019).

In the study area, both population number and population density were increased from 93,163 in 1972 to 556,127 in 2019 mainly due to natural increase and migration. The population density (Figure 9) in the Mekelle city region within 47 years increased from 104 to 620 which is greater than the initial period by 596.15% persons per square kilometer between 1972 and 2019 and higher than the national average in 1972, i.e., 30.14 people per square km. The current population density is much higher than the national population density of Ethiopia which is 109.22 people per $km^2$ as of 2018. This higher population density infers that there is a need for the high demand for the ecosystem, which led to LULC change. Thus, the increasing population density may continue in putting pressure on ecosystem services. The demand for ecosystems to provide benefits/services to the population is growing rapidly.

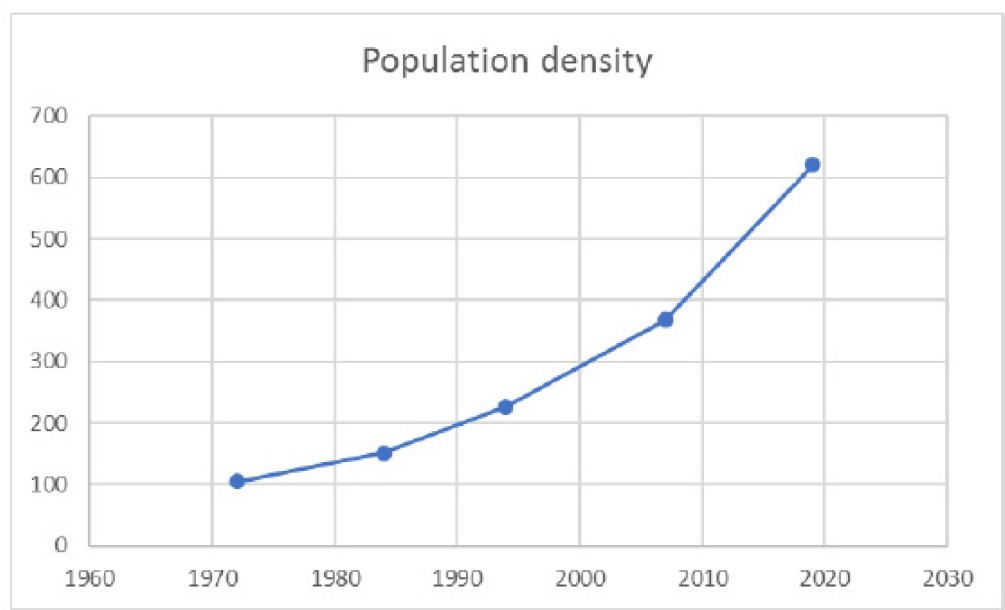

**Figure 9.** Population density of the Mekelle city region.

These official CSA statics showed a clear picture of a high rise in both the rural and urban populations. Both rural and urban populations have increased. Currently, more people are residing in Mekelle city with a population of 423,173 (76%) of the study area, indicating that urbanization of the city region is high. Hence, the urbanization population rise due to natural population growth and migration of the people from remote and rural areas to urban areas.

### 4.1.3. Policies, Institutions, and Legal Frameworks

National policies on economic issues, agricultural strategies, natural resource management, and land tenure reform have affected the LULC changes that occurred in the study area. The policy on urban and rural development is among the key drivers in the modification of the landscapes in the Mekelle city region. Several policies continued to influence land-use practices and had implications for the land users. Identification of past and ongoing land-use patterns can assist urban and regional planners in coping with the uncertainty associated with addressing future development agendas. Since the 1950s, several major policies for national development and environmental conservation have been implemented in Ethiopia [11].

### 4.2. Effect of LULC Changes on ESV

Currently, the use of corresponding coefficient value in the estimation of ecosystem-service values is a word widely applied ([10,18,25]). The valuation of global ecosystem services by Costanza [10] has attracted the attention of few Ethiopian researchers recently. Many of them have been using the methods to assess the ecosystem services for ecosystem degradation induced by LULC. However, it turned out that there are several shortcomings in the direct adaptation of the method. To minimize this problem, this study employed expert FGD.

The value of ecosystem services was obtained by multiplying the relative area of each land cover from 1972–2019. The amount of ESVs changes differed among the nine LULC types of the entire study area in different study years. Among the LULC types, the highest loss was in the water body, riverside vegetation, natural forest, plantation forest, and grassland, consecutively. The dramatic decline in the water ecosystem seriously affected the total ecosystem-service values in the study area during the period 1972–2019. However, the ESVs of bushes and shrubs, and water bodies increased from 2012 onwards. The net changes in ESVS in USD million in the period 1984–1992 were the highest with 106 million losses. However, after 2012, the total ESV of the landscape started to increase by 2.5 million. This study provides a valuable contribution to understanding interactions between LULC changes and ecosystem services supplied by watersheds.

The ESV results indicate that in the last five decades, a decline in the extent of ESV changes arose as a result of LULC dynamics. A study on the water quality of the Tsada Agam river in the core Mekelle city of the study area found that the river water was unfit for human consumption. The inappropriate disposal of industrial effluents and other municipal wastes contributes greatly to the poor quality of river water ([26]). This shows that there are ecosystem service degradations that affect its value. The decline of ecosystem-service values reflected the effects of ecological ruin [2]. LULC variations are prevalent, having impacts on the total ecosystem-service values at the landscape level and bringing about severe changes in ecosystem services [20] Representing the economic value of ecosystem services in the Mekelle city region can make a convincing case for the conservation of the ecosystem to decision-makers.

Across the world, previous findings showed a decline in ecosystem-service values. A study by [27] on the east African rift system RVLR, Ethiopia revealed that approximately USD 196.04 × 106 of ES value was lost from 1986 to 2018. Many other studies showed that LULC types have resulted in a significant loss of ecosystem-service values. On the other hand, an increase of ESVs was observed by [28] in the upper reaches of the Heihe river basin, northwestern China, with the total ESV increasing from USD 1207.33 million (USD) in 2001 to USD 1479.48 million in 2015. The finding is that ESVs vary in different locations. Hence, the effects of land-use changes on ESVs are location-specific. These findings are in agreement with the findings by [10] For the entire biosphere/planet, the value (most of which is outside the market) is estimated to be in the range of USD 16–54 trillion (1012) per year, with an average of USD 33 trillion per year.

The findings of the Mekelle city region show that decreases in ESVs were higher than most previous studies and the global average. The quantification of ecosystem service value

changes provided understandings of the current status of the ecosystem. LULC can affect the ecosystem service. This study shows that there was a correlation between landscape patterns and ESVs. The correlation analysis between implied land-use dynamics had a significant impact on ecosystem-service values. The ecosystem services in the study area contribute to the human welfare of the Mekelle city region and particularly to Mekelle city residents. The study estimated the economic value of 17 ecosystem services for 16 biomes, based on published studies. Global estimates expressed in monetary accounting units are useful to highlight the extent of ecosystem services, and the underlying data and models can be applied at multiple scales to measure the changes resulting from various scenarios and policies [29]. The study results provide a new way to effectively identify the relationship between LULC change and ESV using the Pairwise Granger causality test at the city-region scale.

In Ethiopia, studies on ecosystem-service values at the city-region level are scarce. The study is the first of its kind in Ethiopia cities that can contribute significantly to the environmental sustainability of cities and their hinterlands. Previous studies carried out ESVs on rural areas or urban areas in isolated landscapes. The effects of land-use dynamics in the city region have not been assessed so far. Quantitative assessment of the effects of land-use dynamics on the value of ecosystem services is very crucial. In the long term, maintaining functioning ecosystems is the most cost-effective solution to meeting human needs. Therefore, acknowledging the effects of land-use dynamics on ESVs in the study area is vital to raise the understanding of decision-makers for proper spatial planning.

The study found that focus group discussions are very beneficial to support ESV quantifications. The results of this study demonstrate the value of combining qualitative and quantitative methods to improve the reliability and validity of ecosystem-service values. This study estimated ecosystem service values using the social–ecological approach in the city region. Solving the current ecological crises requires new interdisciplinary and holistic approaches. The study of social–ecological systems focuses on understanding the relationships between nature and society, analyzing the contributions made by ecosystem services to human beings, and investigating the effects of human actions on the ecosystem. Ecosystem-service valuation failures have resulted from a misunderstanding of social-ecological system dynamics, and work of this kind can make a significant contribution to ESV quantification. Hence, it is possible to develop improved policy targets using experts who have different academic backgrounds relevant to the environment, economics, and ecology. This helps in understanding ecosystem service changes between urban and rural environments, relating those changes to societal and climate drivers, and providing science-based tools to inform policy decisions about the sustainable management ecosystem services. This study has a significant role to advance the development of a comprehensive framework that integrates the multidimensional value of ecosystem services.

## 5. Conclusions and Recommendation

The study follows a mixed-methods approach to evaluate the ecosystem services provided by the city region. The qualitative part identified key ecosystem services and provided a quantitative methodology applied to monetarily evaluate the relevant ecosystem services of the study area. The qualitative approach relied on focus group discussions. The local experts described several related ecosystem services. The experts further showed the great potential of ESV for ecological sustainability. To quantify ESVs, the study used the LULC dataset over five decades to estimate the ESV over temporal and spatial scales. To show the changes in the service values along with land-use dynamics, ESVs in the Mekelle city region were quantified from 1972 to 2019.

It was concluded that ESVs varied with land-use change. This study quantified the effects of land-use dynamics of ESVs in the Mekelle city region which is characterized by semi-arid climate, then analyzed and discussed the relationship between ESVs and land-use patterns. The result showed that ESVs increased post-2010 with land-use change due to integrated watershed management. The study concludes that the historical decline

of ESVs showed an environmental severe environmental degradation. This study accepts the hypothesis that LULC changes have an impact on ESV over spatial and temporal scales. This valuation can be used as a basis for policymaking in spatial planning. The study concludes that the decline of ESVs reflected the effects of environmental degradation in the studied landscape and we suggest further studies to explore future options and formulate intervention strategies. In this regard, the following is suggested: (1) Mainstreaming ecosystem-service values/environmental costs of ecosystem service loss into policy and decision-making (2) strengthening ecological restoration to enhance ecosystem-service values (3) Designing intervention strategies to sustain the provision of ecosystem services (4) sustainable land-use planning should be undertaken with an emphasis on riverside vegetation, protection of water bodies and forests and (5) urban planners should minimize the effect of LULC changes on ESVs during the planning process.

Finally, the study seeks to contribute to the growing literature on ecosystem-service valuation. Ethiopia is not an exception to this global LULC trend because modifying land-use patterns is resulting in environmental degradation. The emerging payment for ecosystem services (PES) supported by regional spatial planning should be promoted and expanded at the local level in line with national and international agendas. The authors believe the findings will inform policymakers for regional spatial planning, environmental managers, and the general public on the continuing changes and contribute to developing effective land-use policy in Ethiopia. The outcomes will contribute to an improved understanding of the complex adaptive nature of the city and the social–ecological dynamics in the city region characterized by high population growth, built-up expansion, and a mismatch between high demand and diminishing supply of ecosystem services. Once tested in other city regions with a similar societal and institutional setting, this result could lead to PES and regional policy strategies for rapidly urbanizing cities. This result can be applied locally or regionally in a rapidly urbanizing world. Hence, there is a need for policy change.

**Author Contributions:** (1) S.K.W. makes substantial contributions to conception and design, and/or acquisition of data, and/or analysis and interpretation of data; (2) Co-author K.Y. contribution is on conceptual design, interpretation of data and write up. All authors have read and agreed to the published version of the manuscript.

**Funding:** This research received no external funding.

**Institutional Review Board Statement:** Not applicable.

**Informed Consent Statement:** Not applicable.

**Data Availability Statement:** The dataset(s) supporting the conclusions of this article is (are) available on Google Scholar, JSTOR, Google, and all publicly available datasets are fully referenced in the reference list.

**Acknowledgments:** My thanks go to Tigray Bureau of trade, industry and urban development, Tigray (Ethiopia), the bureau of civil service, and the Ethiopian civil service university for allowing me to pursue this study and for the financial support, without which this achievement would not have been possible. I would like to thank all institutions and individuals at the national, regional, district and city levels who provided me with information for this study.

**Conflicts of Interest:** The authors declare no conflict of interest.

## Abbreviations

The following abbreviations are used in this manuscript:

| | |
|---|---|
| ERDAS | Earth Resource Development Assessment System |
| ESV | Ecosystem-Service Value |
| GTP | Ground Truthing Points |
| ha | Hectare |

| IRS | Thermal Infrared Sensor |
| Landsat | Land Satellite (US Satellite Series) |
| LULC | Land Use/Land Cover |
| MSS | Multispectral Scanner |
| NASA | National Aeronautics and Space Administration |
| OLI | Operational Land Imager |
| R.F | Rainfall |
| TEEB | The Economics of Ecosystems and Biodiversity |
| TM | Thematic Mapper |

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
