# Peer review of "Measuring the Semi-Century Ecosystem-Service Value Variation in Mekelle City Region, Northern Ethiopia"

_sustainability, doi:10.3390/su131810015_

Round 1
Reviewer 1 Report
This paper seems to present extensive use of GIS and sat images to quantify and qualify land use and land cover changes resulting in changes to ecosystem services. I am not expert in this methodology and have typically relied on extensive ground based assessment. Is it possible to really form a coherent argument of such valuation changes relying on remote sensed data? It seems that one of the largest drops in ecosystem service values is presented in your table 4, between 1984 and 1992. I want to know why such as huge drop in this time frame where as the trends during the other years was more consistent. Also what is only lightly touch upon are real recommendations for how to mitigate future degradation. Less than one paragraph at the very end of the paper. As an applied scientist, this is what is perhaps more useful information, policy changes, and not such as dearth of tables and charts of the analysis.
Author Response
Dear reviewers,
First and foremost, we need to thank you for your constructive comments and we are highly motivated by your critical recommendations. We have addressed the comments provided by both reviewers. We have read and incorporated the manuscript review checklist for R1 and R2. In addition to this, we have read and critically considered the comments provided in the manuscript line by line by R1 and R2.
Authors reply to the review report (Reviewer 1)
1.English language and style
For English language the whole manuscript spell was checked using Grammarly and proper spelling has been used.
- Arguments and discussion of findings coherent, balanced and compelling
This was improved bases on the comments provided and the whole article was again revisited line by line.
- Reference
To make citation and references to the standard we have utilized EndNote v.7.1 reference manager software.
- Comments and Suggestions incorporated
This ecosystem services valuation relied on extensive ground-based assessment using remotely sensed data. It is possible to really such valuation changes on remote sensed data. This study focuses at aggregate scale valuation of ecosystem services. In the study area and in other similar location no data exists about previously carried out to value similar services. Adapted ecosystem service value coefficients of the target LULC types was used for this study. In this research study, each of the LULC types of the different reference years was compared with those representative biomes (LULC types) in order to obtain their corresponding ecosystem service value coefficients identified and validating this data by involving experts as to the need for the research from different regional sectors with different professions relevant to the study city region.
The overall gain and loss in ecosystem from the changes for LULC of two time periods 1984-1992 indicates US$ 106 million decline of ecosystem service values was registered. This was corrobated by document review. The document analysis shows that within 1984-1992 severe destruction of ecosystem was observed. Various ecosystems were dramatically reduced. This was further aggravated by mismanagement of natural resources and Severe damage of ecosystem (water body, forests, bushes and shrubs, and river side vegetation). Drought and environmental deterioration have historically imposed heavy costs in Tigrai, Ethiopia and in particular the study area (Esser & Haile, 2002). In 1985 there was heavy drought. Environmental degradation is severe in Mekelle city region which is threatening many parts of the study area(Veen, 2014). The disastrous climatic conditions and civil unrest (War between central government and local political party known as TPLF has significantly contributed to huge drop of ecosystem service values. A study by (Ashenafi, 2014)showed decreasing trends in the annual rainy days (1961–2003).
Recommendations for how to mitigate future degradation.
The study seeks to contribute to the growing literature on ecosystem service valuation. Ethiopia is not an exception to this global LULC trend because modifying land use patterns is resulting an environmental degradation. The emerging payment for ecosystem services (PES) supported by regional spatial planning should be promoted and expanded at the local level in line with national and international agendas. Finally, the authors believe the findings will inform policymakers for regional spatial planning, environmental managers, and the general public on the continuing changes and contribute to developing effective land use policy in Ethiopia. The outcomes will contribute to an improved understanding of the complex adaptive nature of the city and the social-ecological dynamics in the city region characterized by high population growth, built-up expansion, and a mismatch between high demand and diminishing supply of ecosystem services. Once tested in other city regions with a similar societal and institutional setting, this result could lead to PES and regional policy strategies for rapidly urbanizing cities. This result can be applied locally or regionally in a rapidly urbanizing world. Hence, there is the need for policy changes. (Please Refer to page 33 of the manuscript resubmitted)

Reviewer 2 Report
Dear authors,
This research paper describes the important and actual topic – Measuring the Semi-Century Ecosystem Service Values variation in Mekelle city region, Northern Ethiopia. Thus, authors seek to estimate the spatio-tempo ecosystem services value variations. The methodology employed was LULC data sets of remotely-sensed datasets of the year 1972, 1984, 2001,2012 and 2019, ecosystem services value coefficient and expert focus group discussion was used. Authors notice, that the study shows that due to land use changes, the total ecosystem service value is decreasing annually suggesting much more severe ecosystem degradation to occur.
There are some remarks to be noticed: authors present the results of focus group discussions with experts on ESVs of the study very briefly, it seems that some more detailed description of the focus group discussion would be actual seeking the readability of the article; it would be suggested to include to the discussion some more newest theoretical implications, as well.
Author Response
Dear reviewers,
First and foremost, we need to thank you for your constructive comments and we are highly motivated by your critical recommendations. We have addressed the comments provided by both reviewers. We have read and incorporated the manuscript review checklist for R1 and R2. In addition to this, we have read and critically considered the comments provided in the manuscript line by line by R1 and R2.
Authors reply to the review report (Reviewer 2)
1.The content are briefly described and contextualized with previous and present theoretical background and empirical studies related with the science of ecosystem services valuation. And Improvement have been made to the manuscript.
- The research design, questions, hypotheses and methods are clearly stated Please Refer to the manuscript resubmitted in pages:
- This study used explanatory sequential mixed method design (Page 5 about data sources)
- Page 25 in Pairwise Granger Causality Tests
- The arguments and discussion of findings are improved
- The empirical research results are checked again and are clearly presented
- To make citation and references to the standard we have utilized EndNote v.7.1 reference manager software.
- Comments and Suggestions incorporated
Newest theoretical implication of focus group discussion in ecosystem service valuation
The study found that focus group discussions are very beneficial to support ESVs quantifications. The results of this study demonstrate the value of combining qualitative and quantitative methods to improve the reliability and validity of ecosystem service values. This study estimated ecosystem services values using the social-ecological approach in the city region. Solving the current ecological crises requires new interdisciplinary and holistic approaches. The study of social-ecological systems focuses on understanding the relationships between nature and society, analyzing the contributions made by ecosystem services to human beings, and investigating the effects of human actions on the ecosystem. Ecosystem services valuation failures have resulted from a misunderstanding of social-ecological system dynamics, work of this kind can make a significant contribution to ESVS quantification. Hence, it is possible to develop improved policy targets using experts who have different academic background relevant to environment, economics and ecology. Understanding ecosystem service changes between urban and rural environments, relate those changes to societal and climate drivers, and provide science-based tools to inform policy decisions about the sustainable management ecosystem services. This study has significant role to advance the development of a comprehensive framework that integrates the multidimensional value of ecosystem services. (Please refer to discussion part in page 32 discussion part).
Dear reviewers,
First and foremost, we need to thank you for your constructive comments and we are highly motivated by your critical recommendations. We have addressed the comments provided by both reviewers. We have read and incorporated the manuscript review checklist for R1 and R2. In addition to this, we have read and critically considered the comments provided in the manuscript line by line by R1 and R2.
Authors reply to the review report (Reviewer 2)
1.The content are briefly described and contextualized with previous and present theoretical background and empirical studies related with the science of ecosystem services valuation. And Improvement have been made to the manuscript.
- The research design, questions, hypotheses and methods are clearly stated Please Refer to the manuscript resubmitted in pages:
- This study used explanatory sequential mixed method design (Page 5 about data sources)
- Page 25 in Pairwise Granger Causality Tests
- The arguments and discussion of findings are improved
- The empirical research results are checked again and are clearly presented
- To make citation and references to the standard we have utilized EndNote v.7.1 reference manager software.
- Comments and Suggestions incorporated
Newest theoretical implication of focus group discussion in ecosystem service valuation
The study found that focus group discussions are very beneficial to support ESVs quantifications. The results of this study demonstrate the value of combining qualitative and quantitative methods to improve the reliability and validity of ecosystem service values. This study estimated ecosystem services values using the social-ecological approach in the city region. Solving the current ecological crises requires new interdisciplinary and holistic approaches. The study of social-ecological systems focuses on understanding the relationships between nature and society, analyzing the contributions made by ecosystem services to human beings, and investigating the effects of human actions on the ecosystem. Ecosystem services valuation failures have resulted from a misunderstanding of social-ecological system dynamics, work of this kind can make a significant contribution to ESVS quantification. Hence, it is possible to develop improved policy targets using experts who have different academic background relevant to environment, economics and ecology. Understanding ecosystem service changes between urban and rural environments, relate those changes to societal and climate drivers, and provide science-based tools to inform policy decisions about the sustainable management ecosystem services. This study has significant role to advance the development of a comprehensive framework that integrates the multidimensional value of ecosystem services. (Please refer to discussion part in page 32 discussion part).
Dear reviewers,
First and foremost, we need to thank you for your constructive comments and we are highly motivated by your critical recommendations. We have addressed the comments provided by both reviewers. We have read and incorporated the manuscript review checklist for R1 and R2. In addition to this, we have read and critically considered the comments provided in the manuscript line by line by R1 and R2.
Authors reply to the review report (Reviewer 2)
1.The content are briefly described and contextualized with previous and present theoretical background and empirical studies related with the science of ecosystem services valuation. And Improvement have been made to the manuscript.
- The research design, questions, hypotheses and methods are clearly stated Please Refer to the manuscript resubmitted in pages:
- This study used explanatory sequential mixed method design (Page 5 about data sources)
- Page 25 in Pairwise Granger Causality Tests
- The arguments and discussion of findings are improved
- The empirical research results are checked again and are clearly presented
- To make citation and references to the standard we have utilized EndNote v.7.1 reference manager software.
- Comments and Suggestions incorporated
Newest theoretical implication of focus group discussion in ecosystem service valuation
The study found that focus group discussions are very beneficial to support ESVs quantifications. The results of this study demonstrate the value of combining qualitative and quantitative methods to improve the reliability and validity of ecosystem service values. This study estimated ecosystem services values using the social-ecological approach in the city region. Solving the current ecological crises requires new interdisciplinary and holistic approaches. The study of social-ecological systems focuses on understanding the relationships between nature and society, analyzing the contributions made by ecosystem services to human beings, and investigating the effects of human actions on the ecosystem. Ecosystem services valuation failures have resulted from a misunderstanding of social-ecological system dynamics, work of this kind can make a significant contribution to ESVS quantification. Hence, it is possible to develop improved policy targets using experts who have different academic background relevant to environment, economics and ecology. Understanding ecosystem service changes between urban and rural environments, relate those changes to societal and climate drivers, and provide science-based tools to inform policy decisions about the sustainable management ecosystem services. This study has significant role to advance the development of a comprehensive framework that integrates the multidimensional value of ecosystem services. (Please refer to discussion part in page 32 discussion part).
